# Reactive Oxygen Species in Osteoclast Differentiation and Possible Pharmaceutical Targets of ROS-Mediated Osteoclast Diseases

**DOI:** 10.3390/ijms20143576

**Published:** 2019-07-22

**Authors:** Taiwo Samuel Agidigbi, Chaekyun Kim

**Affiliations:** Laboratory of Leukocyte Signaling Research, Department of Pharmacology, Inha University School of Medicine, Incheon 22212, Korea

**Keywords:** reactive oxygen species, osteoclasts, osteoporosis, osteoclast differentiation

## Abstract

Reactive oxygen species (ROS) and free radicals are essential for transmission of cell signals and other physiological functions. However, excessive amounts of ROS can cause cellular imbalance in reduction–oxidation reactions and disrupt normal biological functions, leading to oxidative stress, a condition known to be responsible for the development of several diseases. The biphasic role of ROS in cellular functions has been a target of pharmacological research. Osteoclasts are derived from hematopoietic progenitors in the bone and are essential for skeletal growth and remodeling, for the maintenance of bone architecture throughout lifespan, and for calcium metabolism during bone homeostasis. ROS, including superoxide ion (O_2_^−^) and hydrogen peroxide (H_2_O_2_), are important components that regulate the differentiation of osteoclasts. Under normal physiological conditions, ROS produced by osteoclasts stimulate and facilitate resorption of bone tissue. Thus, elucidating the effects of ROS during osteoclast differentiation is important when studying diseases associated with bone resorption such as osteoporosis. This review examines the effect of ROS on osteoclast differentiation and the efficacy of novel chemical compounds with therapeutic potential for osteoclast related diseases.

## 1. Bone Modeling and Remodeling

The human body undergoes constant bone remodeling, a process that maintains the strength and homeostasis of bones by replacing worn-out bone tissue with newly synthesized calcified matrix. Bone formation and resorption need to be in close equilibrium for normal remodeling of bone [1]. However, in disease conditions, the overall rate of bone resorption (remodeling) exceeds the rate of bone formation (modeling), which results in a decrease in bone mass without a defect in its mineral contents. Bone is deposited by osteoblasts (OBs) in a process called ossification and is degraded by osteoclasts (OCs) [1,2].

OBs and OCs are majorly found in the periosteum and in the endosteum. OBs, the major component of bone from mesenchymal stem cells, secrete matrix proteins and transport mineral into the bone matrix. Prior to organic matrix mineralization, the cells are referred to as osteoids. During bone formation, a large number of OB progenitors expressing *ColIA1/Runx2* are established before the proliferation phase of differentiation. During this period, premature OB progenitors express alkaline phosphatase [2]. The transition of premature OBs to mature OBs is characterized by an increase in the expression of osterix, osteocalcin, bone sialoprotein I/II, and collagen type I. Thereafter, OBs undergo morphological changes, becoming large and active cuboidal cells on the surface layer of the bone matrix and later develop into osteocytes [1,2,3]. Physical activity and chemical signals from osteocytes maintain bone mass and size [1,2,3,4]. Another important step in maintaining bone homeostasis involves the secretion of sclerostin by osteocytes, a protein that inhibits OB activities by inactivating bone turnover and synthesis when osteon reaches a limiting size [3,4,5].

OCs are large multinucleated cells with more than three nuclei that differentiate from mononuclear/macrophage progenitors [1]. Briefly, OCs originate through a process involving multiple steps, starting from the commitment of hematopoietic stem cells into the monocyte-macrophage lineage. Then, premature OCs proliferate into mature OCs, followed by OC polarization induced by integrin αvβ3 which forms a resorptive machinery that facilitates the attachment of OCs to the bone matrix and subsequent formation of a ruffled border that results in the process of resorption [5,6]. OC differentiation from hematopoietic progenitor lineage requires macrophage colony stimulating factor (M-CSF) and receptor activator of nuclear factor kappa-B (NF-κB) ligand (RANKL) to undergo osteoclastic differentiation [1,2,3,4,5,6]. These factors are the key determinants in stimulating the differentiation and activation of OCs, and are very crucial in bone remodeling.

## 2. Mechanism of Bone Resorption Process

OBs and OCs are responsible for a series of events during bone modeling and remodeling. Before the discovery of RANKL, RANK, and OPG, many research groups observed that OBs produce cytokines to regulate OC differentiation and formation. The discovery of OPG/RANKL/RANK pathway provide pieces of evidence that OBs can regulate OCs through cell–cell contact, paracrine factors and interaction via bone matrix [7,8,9]. Initiation of the resorption process involves the OCs gaining access to the bone that is to be resorbed. Some key elements regulating this process are systemic hormones, nerve signals, vascular agents, and growth factors, such as cytokines, chemokines, cell adhesion molecules, molecules of the extracellular matrix, and proteinases. Interestingly, the pro-inflammatory cytokines, such as IL-1 and TNF-α have been implicated in the formation and functional activities of OCs [1,2,3,4,5,6]. The binding of RANKL to its receptor RANK, recruits the adapter molecule, TNF receptor associated factor 6 (TRAF6), resulting in the activation of a signal cascade of various mitogen activated protein kinases (MAPKs), including extracellular signal-regulated kinase (ERK), p38 MAPK, and c-Jun N-terminal kinase (JNK), which in turn activates nuclear factor of activated T cells 1 (NFATc1), a key step in RANKL-induced OC differentiation (Figure 1). OCs convey signals to activate stromal cells thereby leading to the production of OC precursors which later differentiate into mature OCs. These cells resorb bone in resorption lacunae by generating acidic pH gradient between the bone surface and cell, favoring the action of the OC proteinases.

Carbonic anhydrase II (CAII) is the major cytoplasmic source of protons needed for acidification of the lacuna at gradient pH. This enzyme converts carbon dioxide (CO_2_) to carbonic acid (H_2_CO_3_), which subsequently undergoes ionization to form carbonate and hydrogen ions [6,10,11,12]. A proton pump, ATPase transfers the protons produced by CAII into intracellular vesicles. Mitochondria are more abundant in OCs than in any other cell type, thereby generating sufficient ATP needed to generate hydrogen ions. The process of acidification is completed by a membrane potential driven chloride transport exchanger and hydrogen ion is transported along with the ATPase to the ruffle border through endosomes [12,13,14]. The membrane exchangers Na^+^ and Cl^−^ regulate the internal pH at physiologic equilibrium. Cytoplasmic calcium-binding protein, calmodulin concentrated in the OC cytoplasm next to the ruffle border, controls the effects of intracellular calcium and the ATPase across the ruffle border. As resorption activities of OCs progress, the level of cytoplasmic calcium also increases which deactivates the OCs, causing cell detachment from the bone matrix, ultimately leading to a loss of the ruffle border [10,11,12,13].

Lysosomal enzymes, collagenases, and proteinases degrade the exposed organic materials and collagen fiber which are removed and sent into the extracellular space. When the OCs leave the resorption lacuna, phagocytes then engulf the debris and OBs migrate to begin another set of bone formation (modeling) [10,11,12,13]. Cysteine proteases and matrix metalloproteinases (MMPs), such as collagenase stromelysin, continue with the degradation process while calmodulin, an antagonist and MMP inhibitor, can block resorption by inhibiting acidification of the resorptive compartment [10,11,12,13].

## 3. Reactive Oxygen Species (ROS)

ROS include several reactive molecules and free radicals, such as superoxide anion (O_2_^−^), hydrogen peroxide (H_2_O_2_), and hydroxyl radical (HO^●^). These molecules produced at the electron transport chain during aerobic respiration can influence biological functions, such as cell signaling and homeostasis. ROS are necessary for the regulation of cell proliferation, survival, metabolism, apoptosis, differentiation, and migration. Several recent studies have shown that ROS are essential intracellular secondary messengers which carry out many of the normal functions including apoptosis, gene expression, and the activation of cell signaling cascades [15,16,17]. ROS are interesting molecules with dual roles. They can be beneficial when they function as intracellular signaling agent, and are harmful when their levels increase with age or with the onset of an inflammatory state, or with age related diseases including osteoarthritis. All of these subsequently lead to bone destruction and ultimately cell death [18,19,20]. Recent studies have also indicated that ROS are important components which regulate the differentiation process of OCs [21,22].

Bone homeostasis depends on the cross-talk between OBs and OCs. Fundamentally, the balance between formation and resorption has a critical influence on bone mass and strength but start to decline during onset of aging with an increased in OC activity and a declined in OB activity [7]. Co-culture models of OBs and OCs have been used to understand the relationship between OB and OC and effect of ROS in bone development [7,8,9,23,24]. Ohyama et al. [23] has reported that sudachitin, a polymethoxyflavone derived from *Citrus sudachi*, inhibited OC formation in a co-culture of OBs and OCs in the presence of pro-inflammatory factors via suppression of the production of intracellular ROS. Arakaki et al. [25] reported that N-acetyl cysteine (NAC) treatment reduced ROS production as well as bone mineralization during OB differentiation of MC3T3-E1 cells, suggesting that the ROS pathways play a role in OB differentiation. Maria et al. [26] reported that treatment of melatonin, strontium citrate, vitamin D3, and vitamin K2 increased osteoblastogenesis and decreased osteoclastogenesis in co-culture of OBs and OCs. Melatonin-micronutrient osteopenia treatment study was carried out using human OBs and OCs co-culture models to check if melatonin in combination with three other natural bone-protective antioxidants/micronutrients (strontium, vitamin D3, and vitamin K2) could improve health related quality of life in postmenopausal osteopenic women. Strontium increased vertebral and femoral bone density and reduced fracture in both postmenopausal osteopenic and osteoporotic women [26].

### 3.1. Mitochondrial ROS

The mitochondrion is a vital intracellular organelle responsible for energy production and intracellular signaling in the eukaryotic system. Mitochondrial dysfunction often accompanies and contributes to human disease development [27,28]. During the process of oxidative phosphorylation, mitochondria utilize oxygen to generate ATP from organic molecules, but also produce ROS in the same process which affects various signaling processes within the cell. Production of mitochondrial ROS (mtROS) takes place mainly via the electron transport chain on the inner membrane of the mitochondria during the process of oxidative phosphorylation (Figure 2).

Electron leakage at complex I (NADH dehydrogenase) and complex III of the electron transport chain leads to a partial reduction of oxygen to form O_2_^−^, which is simultaneously converted to H_2_O_2_ by superoxide dismutase 2 (SOD2) in the matrix and by SOD 1 in the intermembrane space [29]. Considering the electrophilic properties and half-life of O_2_^−^, it might be unable to pass through the mitochondrial outer membrane and hence unlikely to participate in signal transduction. Rather, O_2_^−^ interacts with NO to form peroxynitrite (ONOO^●−^), a strong oxidant capable of inducing DNA damage, disruption of the mitochondrial architecture, and irreversible protein modification [30].

### 3.2. NADPH Oxidase

NADPH oxidase (NOX), a multicomplex enzyme, has been identified as one of the key sources of ROS. Seven members of NOX family have been identified and characterized including NOX1, NOX2 (gp91^phox^), NOX3, NOX4, NOX5, DUOX1, and DUOX2. Each member has six transmembrane domains together with a cytoplasmic domain, which bind NADPH and FAD and are distinguished by the specific catalytic subunit, interacting proteins, and subcellular localization [31]. After the assemblage, the active complex generates O_2_^−^ by donating an electron from NADPH in the cytosol to oxygen in the extracellular space [22,32].

Although NOX1 is most abundantly expressed in the colon epithelium, it is also expressed in a variety of other cells including vascular smooth muscle cells, OCs, and retinal pericytes [33,34]. There is a high sequence identity of approximately 60% between NOX1 and NOX2 genes [31,35]. The molecular mass of NOX1 was suggested to be in the range of 55–60 kDa [36]. The cytosolic subunits of NOX1 are named as NOXO1 (NOX organizer 1) and NOXA1 (NOX activator 1).

NOX2 is a member of the catalytic subunit of the superoxide-generating NOX of phagocytes. NOX2 has been found in OC precursors and OBs and is highly expressed in macrophages and neutrophils where it generates O_2_^−^ essentially needed for killing of pathogens by respiratory burst [22,37]. After cellular activation, three cytoplasmic subunits—p47^phox^, p67^phox^, and p40^phox^—translocate to the membrane and bind with the membrane-bound flavocytochrome *b_558_* to form active NOX2. Mitochondria are also known to regulate NOX2 activation [38]. mtROS induce slower but steady NOX2 activity, suggesting a cross-link between mitochondria and NOX2. In this review, we summarize the most important features of NOX2, with emphasis on the properties that allow a better understanding of its roles in osteoclastogenesis.

NOX3 is a p22^phox^-dependent plasma membrane enzyme found in many cell types such as human fetal kidney cells, in the spiral ganglion, cochlear sensory cells of the inner ear, and liver cells [39,40]. NOX3 expression stabilizes p22^phox^ protein levels and leads to p22^phox^ translocation from cytosol to the plasma membrane for its activity. NOX3 has uncommon elasticity, and it can be activated by both interaction with p22^phox^ and the organizers p47^phox^ and NOXA1 in the absence of activators [39,40].

Nox4 is highly expressed in renal and endothelial cells [41,42,43], but is also present in many other cell types and hence may have a broader cellular function than other NOX isoforms [32,33,34,35,36,37,38,39,40,41,42]. NOX4 is constitutively active and generates H_2_O_2_ independent of cytosolic activator protein in parallel to NOX1–NOX5 [42,43,44,45]. Due to the common regulatory subcomponent (p22^phox^) that NOX4 shares with NOX1 and NOX2. NOX4 is believed to be constitutively active and responsible for the production of basal ROS [45].

NOX5 was reported in the testis, lymph, spleen, bone marrow, and various other tissues [44,45,46]. NOX5 is the most divergent NOX isoform and possesses a unique NH_2_ terminal that contains four Ca^2+^ binding sites responsible for enzyme activation [45,47]. There are five known variants of NOX5 that have been described so far, namely -α, -β, -δ, -γ and a truncated variant -S or -ε [44]. NOX5 -α and -β variants are active and produce ROS. However, the activity of the δ- and γ- isoforms has not yet been reported [44].

DUOX1 and DUOX2 are also members of the NOX family with a seventh transmembrane domain and share similar features with other NOX isoforms. Both DUOX1 and DUOX2 have a molecular weight of approximately 180 kDa and were first identified in the thyroid gland as the only source of H_2_O_2_ found in the thyroid follicle [48]. The discovery of DUOX1/2 in thyroid hormone biosynthesis has prompted different research groups to explore the possible role of DUOX in the immune system as a source of extracellular H_2_O_2._

### 3.3. Hydrogen Peroxide

Hydrogen peroxide elicits multiple effects across diverse physiological responses, including cell differentiation, proliferation, and migration [35,49]. H_2_O_2_ is electrophobic in nature and much more stable than O_2_^−^. Additionally, the concentration of H_2_O_2_ in mitochondria is 100 times greater than that of O_2_^−^ [50]. H_2_O_2_ increases after tissue damage and is thought to act as a signal for white blood cells to converge on the site and initiate the healing process. For example, H_2_O_2_ induced the activation of NF-κB/Rel family [51], which is a well-known redox sensitive transcription factor and a major regulatory molecule involved in the regulation of many transcription genes during inflammation. H_2_O_2_ is involved during development by triggering apoptosis and cell proliferation [51,52,53].

## 4. ROS in OC Differentiation and Activity

### 4.1. mtROS and OCs

Mitochondrial ROS are essential for hypoxic enhancement of OC differentiation [54]. In a hypoxic environment, accumulation of mtROS in OCs has been reported and the pathway was suggested to be mediated by mitochondrial respiratory stress signaling induced by hypoxic release of Ca^2+^ from the ER [54,55,56]. Effect of mtROS on OC differentiation and resorption in hypoxic environment was reversed by the mitochondria-specific antioxidant MitoQ [55], which prevented the hypoxic induction of NF-κB and calcineurin-NFAT pathway, while NAC, ascorbate and 1, 2-bis (2 aminophenoxy) ethane N, N, N′, N′-tetra acetic acid (BAPTA) inhibited OC formation at high concentration [55,57].

### 4.2. NOX1 in OCs

NOX1 protein was detectable at a low level in bone marrow macrophages (BMM), but the protein expression of the other NOX members was not detectable except NOX2, which is the main isoform in BMM [34]. It has also been reported that silencing of NOX1 in BMMs significantly decreased ROS production and inhibited OC differentiation [34]. Lee et al. [58] reported that genistein inhibited OC formation by inhibiting the translation and activation of NOX1. NOX mRNA levels were also assessed during the process of OC differentiation to evaluate its contribution in the production of ROS. Silencing of NOX2 by siRNA did not affect RANKL mediated ROS production or OC formation, but on the other hand, NOX1 knockout greatly reduced ROS production and OC differentiation [58]. This result is in contrast with the findings of Xu and colleagues [59], who reported that knockdown of NOX1 had no significant effect on RANKL mediated ROS production, but a reduction was observed when both NOX1 and NOX2 were knocked down [59]. The results suggest that the loss of NOX1 might be compensated by other NOX isoforms during the early stages of OC differentiation and that loss of either of the isoforms is not detrimental to osteoclastogenesis [59].

### 4.3. NOX2 in OCs

We have reported that NOX2-derived O_2_^−^ enhanced RANKL-induced NFATc1 expression in OC signaling [60]. OCs from NOX2 knockout mice showed reduced NFATc1 expression during RANKL stimulation. However, there was no significant difference in NF-κB levels. Recently, it has been reported that cellular NOX2 level is modulated by a negative regulator of ROS (NRROS) by degrading NOX2 dependent ER associated pathway [61]. NRROS increased during OC differentiation and suppressed NOX2 expression, which subsequently masked RANKL-induced OC differentiation [61]. These reports suggest that NOX2 expression and NOX2 derived ROS are closely controlled by negative regulatory frameworks.

### 4.4. NOX4 in OCs

NOX4 is expressed at a higher level in OCs than in precursor cells which suggested that NOX4 is upregulated during the differentiation and development of OCs [62]. The abrogation of the standard level of NOX4 expression in OCs inhibits both the production of O_2_^−^ and subsequent bone resorption activity [62,63]. NOX4 was not found in bone marrow macrophages, but mRNA expression was detected in OCs after RANKL induction [62]. This finding corresponded with another study which reported that bone marrow has an infinitesimal amount of NOX4 mRNA [59]. Upregulation of NOX4 suggests its physiological importance in the development and formation of OCs [64]. NOX4 knockout mice demonstrated lower OC number and reduced expression of circulating markers of bone resorption [64]. NOX4 is the only NOX family member found in human chondrocytes and is suggested to be a key player involved in cartilage degradation and development of osteoarthritis [65].

### 4.5. H_2_O_2_ and Other ROS in OCs

SOD, NAC, and diphenyleneiodonium (DPI) inhibited OC formation [60], supporting the view that ROS plays an important role in the differentiation of OCs. H_2_O_2_ treatment rescued the defect in OC differentiation in NOX2 knockout mice [60]. In another independent study, DPI treatment negated RANKL-mediated increase in ROS and reduced the total number of OCs formed during the process of differentiation [34]. Additionally, NAC was reported to block RANKL-induced ROS production, MAPKs activation, and osteoclastogenesis. However, activation of MAPKs was observed when cells were treated with H_2_O_2_ [34]. In addition, exogenous H_2_O_2_ induced osteoclastogenesis, indicating that oxidative stress plays a role in the regulation of osteoclastogenesis from both within the cytoplasm and extracellular milieu. NOX/ROS activity in OC differentiation is summarized in Table 1. 

## 5. Regulation of OC Differentiation by Redox Modulation

The antioxidant activity of glutathione is associated with chemo-preventive and therapeutic applications in clinical conditions, such as age-related diseases, cancer, liver cirrhosis, diabetes, and neurodegenerative diseases [66,67,68,69]. The regulation of the balance between reduced glutathione (GSH) and oxidized glutathione (GSSG) has been used as an indicator to determine redox status in cellular milieu [70].

The regulation of redox status during OC differentiation plays an important role in maintaining bone homeostasis and contributes to the understanding of the pathogenesis and therapy of bone diseases, such as osteoporosis and osteopetrosis [71]. GSH and its inhibitor L-buthionine-sulfoximine (BSO) have been implicated in modulation of redox status during OC differentiation [70,71,72,73]. GSH enhanced OC differentiation while BSO reduced it. Pre-treatment of RAW 264.7 cells with BSO depleted total GSH content and inhibited OC differentiation [70]. However, repletion with NAC and GSH restored tartrate resistant acid phosphatase (TRAP) activity and increased the number of OCs compared to control [70,72]. Our recent research [74] also showed that GSH increased OC differentiation while BSO reduced it. Our findings corresponded with a recent study by Fujita and colleagues [75], which reported that GSH increased OC differentiation. However, further research is required to confirm this finding since there have been contradictory reports from other researchers regarding the role of GSH and BSO during OC differentiation [71,76].

## 6. ROS in Signal Cascades of Osteoclastogenesis

RANKL is an important cytokine in osteoclastogenesis, and various intracellular signaling molecules, such as NFATc1, MAPKs, TRAF, and AKT have been reported to regulate OC differentiation [77,78]. TRAF6 plays a key linkage role in ROS production by RANKL and a dominant interfering mutant form of TRAF6, significantly decreased ROS induction, but TRAF6 itself does not directly participate in ROS production [21,34,79]. Rac is a cytosolic component of NOX complex and a downstream signal messenger of the Rho GTPase family, which has been reported to be involved in cytoskeletal organization and is responsible for the activation of NOX. The expression of a dominant-negative mutant of Rac1 blocked ROS production indicating that Rac1 is responsible for regulating the generation of ROS during OC differentiation [79]. RANK, TRAF6, Rac1, and NOX might form the sequential order of the signaling cascade of ROS production. Although the target molecules of ROS in RANKL signaling cascade remain canonical; nevertheless, several reports have suggested that activation of MAPKs, PI3K, and NF-κB are the main downstream events [53,80]. Additionally, Kim et al. [81] reported that NFATc1 is the master switch for OC differentiation. We measured the expression levels of NF-κB, AP-1, and NFATc1 in NOX2 knockout mice during RANKL induced OC differentiation. Our findings agree with earlier reports which indicated that NFATc1 signaling could be the crucial downstream signaling event in RANKL-mediated ROS signaling [81].

## 7. ROS as Pharmacological Targets for OC Associated Diseases

OCs are essential for bone renewal/regeneration but increase in their cellular activities have been associated with bone diseases such as osteoporosis, rheumatoid arthritis, and osteoarthritis [82]. The detrimental effects of ROS in damaging macromolecules and causing cellular stress are evident. Understanding the role of ROS in OC signaling cascade in relation to other cells including osteocytes and OBs could lead to the development of novel therapeutics that target ROS and halt excessive bone resorption.

Apocynin, a ROS scavenger inhibited downstream NOX isoforms and ROS formation [83]. DPI, a non-reversible inhibitor of flavoprotein, NOX, xanthine oxidase, and mitochondrial electron transport chain have also been reported to reduce OC formation [83]. Ewha-18278, a pyrazole derivative, protected ovariectomy-induced osteoporosis through the inhibition of NOX and ROS suppression [84]. The potential effect of Ewha-18278 as a novel anti-osteoporotic agent was assessed in ovariectomy-induced osteoporotic ddY mice. It was orally applied once a day for 28 days. Oral administration of Ewha-18278 (5-20 mg/kg/day) led to recovery in bone parameters, resulting in increased bone strength. Besides, the number of OBs was increased in Ewha-18278 treated tibia and femur of mice compared to ovariectomy control mice [84].

ML171 and thioridazine are potent antagonists and NOX inhibitors, potentially scavenging for ROS. Thioridazine is known to be a non-competitive inhibitor of NOX2 [85]. In a case report, dried plum polyphenols, containing antioxidant and anti-inflammatory properties, administered to rat models of osteoporosis resulted in attenuated bone loss compared to controls [86]. Simvastatin, a 3 hydroxy-3 methylglutaryl coenzyme A (HMG-CoA) reductase inhibitor, used for the treatment of hypercholesterolemia, led to a reduction in ROS levels and downregulation of osteoclastogenesis signaling cascades. Simvastatin is known to be one of the systemic drugs used to suppress bone resorption by inhibiting osteoclastogenesis through ROS inhibition [87]. The effect of simvastatin on bone mineral density in a total of 69 patients with type 2 diabetes was studied by Chung and colleagues [88]. After 15 months of simvastatin treatment, the treated subjects showed increase in bone mineral density. This study indicated that simvastatin reduces the risk of bone fracture [88]. Peroxynitrites GKT136901 and GKT137831 at sub-micromolar concentrations have been reported to inhibit NOX [83,89,90]. Alliin (S-allyl-l-cysteine sulfoxide), an essential component in garlic extract, scavenged ROS through inhibiting NOX1, c-Fos, and NFATc1 signaling pathways [91].

Katsumata et al. [92] reported that a single local injection of epigallocatechin gallate (EGCG) attenuated bone resorption and orthodontic tooth movement in mice, and EGCG increased the formation of mineralized bone nodules in human osteoblast-like cells [93]. Yun et al. [94] also reported that, EGCG reduced MMP-9 and OC formation. Studies have provided a primary support for nutritional approaches to antioxidant strategies for the prevention of bone loss [95,96]. Rao et al. [97] reported that a cross-sectional study of 33 postmenopausal women showed that high lycopene consumption was associated with a moderate serum level of N-telopeptide of collagen type 1 and lower serum protein oxidation, suggesting a decrease in ROS-induced bone resorption. In another clinical postmenopausal osteoporosis study [98], lycopene-supplementation significantly increased serum lycopene. These findings suggest that lycopene can be administered to exert its potent antioxidant properties in reducing the risk of osteoporosis [98].

The compounds listed in Table 2 have been reported to improve bone homeostasis in animal models and human, they can be locally and systemically applied and can also be administered as additive/supplement to curb ROS-OC diseases. Such intervention will eliminate adverse drug effects associated with the treatment of bone diseases by conventional drugs and/or reduce the financial burden of surgery and hormone replacement therapy. However, intervention studies of ROS therapy in established disease cases are always inconclusive and the quality of the findings are still poor [99].

## 8. Conclusions

In this review, we have summarized the current information regarding the role of ROS in OC differentiation and the therapeutic potential of chemical compounds targeting OC associated diseases. Convincingly, ROS greatly regulates OC differentiation and modulates bone homoeostasis. ROS elicits multiple effects across diverse physiological responses, including bone resorption and differentiation, proliferation, and survival of OCs. Antioxidant pharmacological compounds have been reported to play a crucial role in the management of cellular stress arising from the detrimental effects of ROS in bone diseases. Targeting ROS and NOX isoforms individually rather than a single entity could lead to the development of novel therapeutics for bone related diseases and this will also prevent undue interference with other normal biological activities and functions.

## Figures and Tables

**Figure 1 ijms-20-03576-f001:**
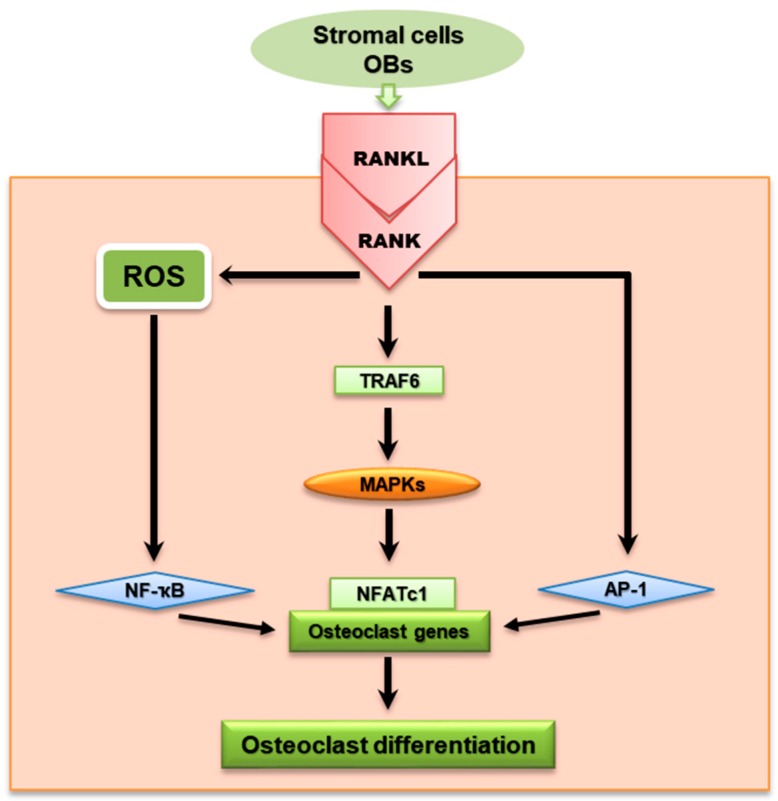
RANKL/RANK signaling pathway from stromal cells/OBs showing downward signaling molecules to the transcription factors in the nucleus that regulate OC differentiation. RANKL is an important cytokine in the OC signal pathway together with various intracellular signaling molecules—such as TRAF, MAPKs, NF-κB, and NFATc1—which are essential for regulating OC differentiation.

**Figure 2 ijms-20-03576-f002:**
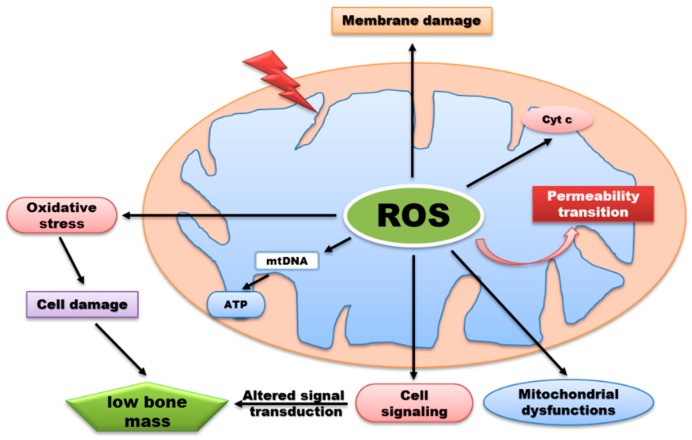
ROS generation by mitochondria can influence various biological functions but excessive production of ROS lead to signaling alteration, membrane damage, release of cytochrome c, and oxidative damage to mitochondrial proteins and DNA. Consequently, oxidative damage can impair synthesis of ATP thereby preventing normal metabolic functions which contributes to a wide range of disease development.

**Table 1 ijms-20-03576-t001:** ROS in osteoclasts.

ROS	Cell Type	References
O_2_**^−^**	RAW 264.7	[61,80,100]
	mBMM	[80,100]
	HD-11EM	[101]
	HL-60	[100,102]
	hBMM	[103]
	hMSC	[104]
	THP-1	[100]
	U937	[102]
	HP100-1	[102]
H_2_O_2_	RAW 264.7	[34,54,55,56,57,58,87]
	HD-11EM	[101]
	hBMM/hPB	[103]
	mBMDM	[34,60,87]
	HL-60	[100,102]
	U937	[102]
	HP100-1	[102]
	THP-1	[100]
	hMSC	[104]
mtROS	RAW 264.7	[54,55,56,57]
	hMSC/hPB	[15,56]
NOX1	mBMM/hBMM	[34,58,59,60,83,84]
	RAW 264.7	[54,58,91]
NOX2	mBMM	[59,60,62,83,84,105]
	RAW 264.7	[62,80,100]
NOX4	mBMM	[49,59,63,84]
	RAW 264.7	[63,64]

**Table 2 ijms-20-03576-t002:** ROS/NOX inhibitors: Basic mechanisms and treatment indication.

Compounds	Diseases	Basic Mechanisms/Treatment Indication	References
Alliin	Osteopenia	NOX/ROS inhibitor	[91]
Alpha-lipoic acid	Collagen-induced arthritis	Reduce ROS	[106]
Antioxidant enzymes	Osteoarthritis	NOX/ROS scavenger	[107]
Antioxidants (DPI, SOD, Peroxidase)	Osteoporosis, osteopenia, osteoarthritis	NOX/NOS/ROS inhibitor Flavoprotein inhibitor	[83]
Apocynin	Bone marrow senescence, osteoporosis	NOX/ROS scavenger	[83,108]
Epigallocatechin gallate	Orthodontic tooth movement	Reduce ROS	[92,93,94]
Extract of dried plum polyphenols	Osteoporosis	ROS and NFATc1 inhibitor NOX/ROS scavenger	[86]
EWHA-18278	Osteoporosis/osteopenia	NOX/ROS inhibitor	[84]
Fermented oyster extract	Bone loss-related diseases	Reduce ROS	[109]
GKT136901, GKT137831	Osteoporosis	NOX/ROS inhibitor	[83,89,90]
Lycopene	Postmenopausal bone loss	Reduce ROS	[97]
ML171/Thioridazine		NOX inhibitor	[85]
Simvastatin	Rheumatoid arthritis	Reduce ROS, AKT, and MAPKs Promote bone formation	[87,88]

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
