# Peer review of "Reactive Oxygen Species in Osteoclast Differentiation and Possible Pharmaceutical Targets of ROS-Mediated Osteoclast Diseases"

_ijms, 2019, doi:10.3390/ijms20143576_

Reviewer 1 Report

This is a well-written review describing several effects of Reactive Oxygen Species on osteoclast differentiation and activity. It is a very complete paper with an interesting section regarding possible pharmaceutical approaches for osteoclast diseases. 

Author Response

Thank you for the kind comment and encouragement.

Reviewer 2 Report

The manuscript by Agidigbi and Kim provides a comprehensive overview on the role of reactive  oxygen species in osteoclast. The most relevant finding on the effects of ROS on osteclast differentiation  and funtionality are discussed with great enphasis on molecular mechanism. Potentials and limitations of ROS as terapheutic targets are aslo discussed.
 However, due to the complex mechanism of bone resorption process and to the dynamic interplay between osteoblasts, osteoclasts, authors should also provide a brief comment on the effect of ROS. Studies providing data from co-colture experiment should be also reported and discussed.

Author Response

The manuscript by Agidigbi and Kim provides a comprehensive overview on the role of reactive oxygen species in osteoclast. The most relevant finding on the effects of ROS on osteoclast differentiation and functionality are discussed with great emphasis on molecular mechanism. Potentials and limitations of ROS as therapeutic targets are also discussed.
However, due to the complex mechanism of bone resorption process and to the dynamic interplay between osteoblasts, osteoclasts, authors should also provide a brief comment on the effect of ROS. Studies providing data from co-culture experiment should be also reported and discussed.

R: We have addressed the interplay between OBs and OCs in page 2 (lines 57-61), and a brief comment on the effect of ROS on co-culture of OBs-OCs experiment has been addressed in page 3-4 (lines 112-128).

Reviewer 3 Report

This is a very good review on the role of ROS in osteoclast Differentiation and possible pharmaceutical targets of ROS-mediated osteoclast diseases. 

The review is clearly and concisely written with good structure and a good running theme throughout the manuscript. 

Overall, there were excellent details on the mechanism of bone resorption and ROS generation in cells and tissue and how altered ROS levels can possibly lead to orthopaedic bone loss in diseases such as Osteoarthritis, Osteoporosis, and Rheumatoid Arthritis. 

However, I felt that in the final part of the review it lacked enough current information on the possible pharmaceutical targets of ROS-mediated Osteoclast Diseases. The authors provide two tables, one for the ROS in osteoclasts and the other of ROS/NOX inhibitors for basic mechanisms and treatments. These are all well good but there should be some discussion in depth as to what some of the major studies, experiments have achieved and where treatments have not been as successful as first predicted.   

As ROS have multifactorial mechanistic roles in cell function and metabolism, how would therapeutic interventions of ROS modulation in disease states not affect normal functions of tissue and organs? How will interventions to ROS-osteoclasts be administered - locally or systematically? It would be more informative in the review if the authors discussed how the therapies might be applied and the possible advantages and disadvantages pertaining to these interventions. 

The authors have limited their review to major diseases such OA, OP, and osteopenia but what about other bone diseases such as Paget's disease or what happens if ROS inhibitors are applied to a patient that has a fracture or multiple fractures.  Does inhibition of ROS affect the natural healing of bone or could it actually affect the natural homeostasis of healthy native bone?

Minor comments

Page 2 Line 44 spelling of sclerostin is wrong.

Page 2 Line 51  Should read "...subsequent formation of a ruffled border..."

Figure 1 what cell type does the large box represent, Osteoclast, osteoclast precursor, monocytes/macrophages?

A number of the references have double number indicators e.g.

88. 88. Chen, C.T.; Shih, Y.R.; Kuo, T.K.; Lee, O.K.; Wei, Y.H. Coordinated changes of mitochondrial biogenesis and antioxidant enzymes during osteogenic differentiation of human mesenchymal stem cells. Stem Cells 2008, 26, 960– 968.

Author Response

However, I felt that in the final part of the review it lacked enough current information on the possible pharmaceutical targets of ROS-mediated Osteoclast Diseases. The authors provide two tables, one for the ROS in osteoclasts and the other of ROS/NOX inhibitors for basic mechanisms and treatments. These are all well good but there should be some discussion in depth as to what some of the major studies, experiments have achieved and where treatments have not been as successful as first predicted.   

R: We have discussed some of the major studies in-depth in page 8 (lines 306-310, lines 318-321) and added more information in page 8 (lines 325-335). In addition, we added more compounds in Table 2.

 As ROS have multifactorial mechanistic roles in cell function and metabolism, how would therapeutic interventions of ROS modulation in disease states not affect normal functions of tissue and organs?

R: We have addressed that question in conclusion (page 10, lines 351-353).

 How will interventions to ROS-osteoclasts be administered - locally or systematically? It would be more informative in the review if the authors discussed how the therapies might be applied and the possible advantages and disadvantages pertaining to these interventions. 

R: We have discussed how the intervention can be applied, possible advantages and limitations in page 8 (lines 336-342).

 The authors have limited their review to major diseases such OA, OP, and osteopenia but what about other bone diseases such as Paget's disease or what happens if ROS inhibitors are applied to a patient that has a fracture or multiple fractures. Does inhibition of ROS affect the natural healing of bone or could it actually affect the natural homeostasis of healthy native bone?

 R: We have included more findings on OC-mediated diseases such as orthodontic tooth movement, rheumatoid arthritis and collagen-induced arthritis in page 9 (Table 2). 

In many cases reported about ROS inhibitor in animal studies, inhibition of OC is always characterized with improvement in bone parameters such as bone volume to total volume, bone mineral density, cortical bone strength and trabecular number. These are indicators that ROS inhibitor can reduce the risk of osteoporotic fractures and enhance natural homeostasis of healthy bone. We added a case study in page 4 (lines 123-128), and in page 8 (lines 306-310).

 Minor comments

R: All minor comments have been corrected.

Page 2 Line 44 spelling of sclerostin is wrong.

R: Corrected.

Page 2 Line 51 Should read "...subsequent formation of a ruffled border..."

R: Corrected as reviewer suggested.

Figure 1 what cell type does the large box represent, Osteoclast, osteoclast precursor, monocytes/macrophages?

R: The box represents ‘schematic diagram from osteoclast precursor down to osteoclast differentiation’.

A number of the references have double number indicators e.g. 88. 88. Chen, C.T.; Shih, Y.R.; Kuo, T.K.; Lee, O.K.; Wei, Y.H. Coordinated changes of mitochondrial biogenesis and antioxidant enzymes during osteogenic differentiation of human mesenchymal stem cells. Stem Cells 2008, 26, 960– 968.

R: All references with double number indicators have been corrected.